# Adaptive Topological Feature via Persistent Homology: Filtration Learning for Point Clouds

**Naoki Nishikawa**
The University of Tokyo
nishikawa-naoki259@g.ecc.u-tokyo.ac.jp

**Yuichi Ike**
Kyushu University
ike@imi.kyushu-u.ac.jp

**Kenji Yamanishi**
The University of Tokyo
yamanishi@mist.i.u-tokyo.ac.jp

## Abstract

Machine learning for point clouds has been attracting much attention, with many applications in various fields, such as shape recognition and material science. For enhancing the accuracy of such machine learning methods, it is often effective to incorporate global topological features, which are typically extracted by persistent homology. In the calculation of persistent homology for a point cloud, we choose a filtration for the point cloud, an increasing sequence of spaces. Since the performance of machine learning methods combined with persistent homology is highly affected by the choice of a filtration, we need to tune it depending on data and tasks. In this paper, we propose a framework that learns a filtration adaptively with the use of neural networks. In order to make the resulting persistent homology isometry-invariant, we develop a neural network architecture with such invariance. Additionally, we show a theoretical result on a finite-dimensional approximation of filtration functions, which justifies the proposed network architecture. Experimental results demonstrated the efficacy of our framework in several classification tasks.

## 1 Introduction

Analysis of point clouds (finite point sets) has been increasing its importance, with many applications in various fields such as shape recognition, material science, and pharmacology. Despite their importance, point cloud data were difficult to deal with by machine learning, in particular, neural networks. However, thanks to the recent development, there have appeared several neural network architectures for point cloud data, such as DeepSet (Zaheer et al., 2017), PointNet (Qi et al., 2017a), PointNet++ (Qi et al., 2017b), and PointMLP (Ma et al., 2022). These architectures have shown high accuracy in tasks such as shape classification and segmentation.

In point cloud analysis, topological global information, such as connectivity, the existence of holes and cavities, is known to be beneficial for many tasks (see, for example, (Hiraoka et al., 2016; Kovacev-Nikolic et al., 2016)). One way to incorporate topological information into machine learning is to use persistent homology, which is a central tool in topological data analysis and attracting much attention recently. By defining an increasing sequence of spaces, called a *filtration*, from a point cloud $X \subset \mathbb{R}^d$, we can compute its persistent homology, which reflects the topology of $X$. Topological features obtained through persistent homology are then combined with neural network methods and enhance the accuracy of many types of tasks, such as point cloud segmentation (Liu et al., 2022), surface classification (Zeppelzauer et al., 2016), and fingerprint classification (Giansiracusa et al., 2019).

37th Conference on Neural Information Processing Systems (NeurIPS 2023).

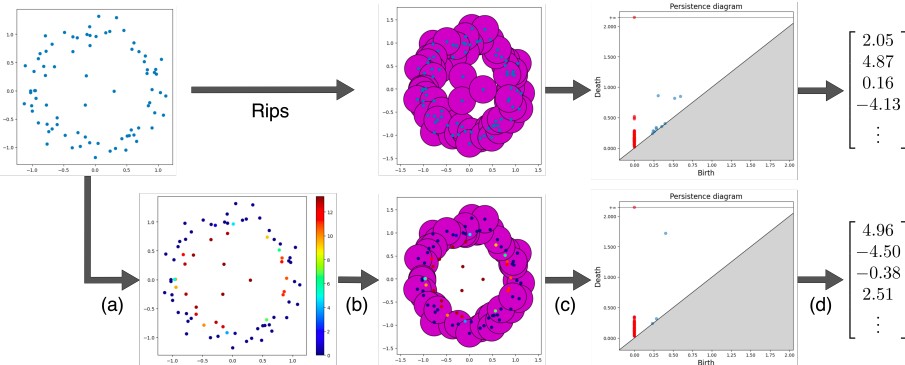

Figure 1: Procedure to utilize our framework to a classification task and its comparison to when we use Rips filtration. For a given set of points, define a value called "weight" for each point in (a), using a neural network method. In (b), a ball centered at each point is gradually expanded. The balls centered at points with larger weights expand later than the balls centered at points with smaller weights. In (c), we examine the topology of the sum of the balls as they grow larger, and aggregate the information into a diagram called a persistence diagram. In (d), the persistence diagram is converted to a vector, which can be used as input of machine learning models. When we use standard Rips filtration, the balls expand uniformly from all of the points, so that we cannot correctly capture the large global hole. On the other hand, when we use our method, the suitable filtration to solve a classification task is chosen, and the weights on the outliers get larger. As a result, we can capture the large global hole in the point cloud. Note that the resulting weights assigned to each point are *not pre-specified, but are fully learned from the data.*

The standard pipeline to use persistent homology in combination with machine learning is split into the following:

(i) define a filtration for data and compute the persistent homology;

(ii) vectorize the persistent homology;

(iii) input it into a machine learning model.

In (i), one needs to construct appropriate filtration depending on a given dataset and a task. A standard way to define a filtration is to consider the union of balls with centers at points in $X \subset \mathbb{R}^d$ with a uniform radius, i.e., take the union $S_t = \bigcup_{x \in X} B(x;t), B(x;t) := \{y \in \mathbb{R}^d \mid \|y - x\| \leq t\}$ and consider the increasing sequence $(S_t)_t$. Regarding (ii), one also has to choose a vectorization method depending on the dataset and the task. For example, one can use typical vectorization methods such as persistence landscape and persistence image. As for (iii), we can use any machine learning models such as linear models and neural networks, and the models can be tuned with the loss function depending on a specific task.

Recently, several studies have proposed ideas to learn not only (iii) but also (i) and (ii). Regarding (ii), Hofer et al. (2017) and Carrière et al. (2020), for example, have proposed methods to learn vectorization of persistence homology based on data and tasks. Furthermore, for (i), Hofer et al. (2020) have proposed a learnable filtration architecture for graphs, which gives a filtration adaptive to a given dataset. However, this method heavily depends on properties specific to graph data. To establish a method to learn filtration for point clouds, we need to develop a learnable filtration architecture for a point cloud incorporating the pairwise distance information. In this paper, we tackle this challenging problem to adaptively choose a filtration for point clouds.

## 1.1 Contributions

In this paper, we propose a novel framework to obtain topological features of point clouds based on persistent homology. To this end, we employ a *weighted filtration*, whose idea is to take the union of balls with different radii depending on points. Choosing a *weight function* $w \colon X \to \mathbb{R}$, we can define

a filtration $(S_t)_t$ by

$$S_t = \bigcup_{x \in X} B(x; t - w(x)).$$

This type of filtration often extracts more informative topological features compared to non-weighted filtrations. Indeed, a weighted filtration plays an important role in various practical applications (Anai et al., 2020; Nakamura et al., 2015; Hiraoka et al., 2016; Obayashi et al., 2022). Then, we introduce a neural network architecture to learn a map that associates a weight function $f(X, \cdot)$ with a point cloud $X \subset \mathbb{R}^d$.

The desirable property for network $f$ is *isometry-invariance*, i.e., $f(TX, Tx) = f(X, x)$ for any isometric transformation $T$ of $\mathbb{R}^d$, so that the associated weighted filtration is isometry-invariant. To address this issue, we implement a function $f$ with a network architecture based on distance matrices. Then, the resulting weight function satisfies the isometry-invariance *rigorously*. Additionally, we show a theoretical result about the approximation ability for a wide class of weight functions, which supports the validity of our architecture.

Our main contributions are summarized as follows:

1. We propose a novel framework to obtain adaptive topological features for point clouds based on persistent homology. To this end, we introduce an isometry-invariant network architecture for a weight function and propose a way to learn a weighted filtration.

2. We theoretically show that any continuous weight function can be approximated by the composite of two continuous functions which factor through a finite-dimensional space. This theoretical result motivates our architecture.

3. We conducted experiments on public datasets and show that the topological features obtained by our method improved accuracy in classification tasks.

Figure 1 presents one typical use of our framework for a classification task. Our method first calculates the weight of each point using a neural network model. The colored point clouds in the second row depict the assigned weights for each point. In the bottom one, our architecture assigns large weights on the outliers in the point cloud, which implies that our method can choose the filtration adaptively to data and tasks. Although the DTM filtration is also effective in the example shown in Figure 1, our method can learn a filtration appropriately for a given task and data in a supervised way. Therefore, our method would be effective for data and tasks with difficulties other than outliers. In §5, we will show that our method improves the classification accuracy compared to the Rips or DTM filtration in several experiments.

## 1.2 Related work

A neural network architecture for finite point sets was first proposed by Zaheer et al. (2017), as DeepSets. Later, there appeared several geometric neural networks for point clouds, such as Point-Net (Qi et al., 2017a), PointNet++ (Qi et al., 2017b), and PointMLP (Ma et al., 2022). These studies addressed the rotation invariance by approximating rotation matrices by a neural network. (Xu et al., 2021) used Gram matrices to implement an isometry-invariant neural network for point clouds. We instead use distance matrices for our isometry-invariant network architecture since the pairwise distance information is important in persistent homology.

Regarding vectorization of persistent homology, in the past, it has usually been chosen in unsupervised situations, as seen in Bubenik and Kim (2007); Chung et al. (2009); Bubenik et al. (2015); Adams et al. (2017); Kusano et al. (2017). However, recent studies such as Hofer et al. (2017) and Carrière et al. (2020) have proposed end-to-end vectorization methods using supervised learning. In particular, Carrière et al. (2020) used the DeepSets architecture Zaheer et al. (2017), whose input is a finite (multi)set, not a vector, for vectorization of persistent homology to propose PersLay. Although our method also depends on DeepSets, it should be distinguished from PersLay; our method uses it for learning *a weight for computing persistent homology* while PersLay learns *a vectorization method of persistence homology*.

For graph data, Hofer et al. (2020) proposed the idea of filtration learning in supervised way, i.e., to learn the process in (i), and this idea was followed by Horn et al. (2021) and Zhang et al. (2022). They

introduced a parametrized filtration with the use of graph neural networks and learned it according to a task. The persistent homology obtained by the learned filtration could capture the global structure of the graph, and in fact, they achieved high performance on the graph classification problem. However, filtrations for point clouds cannot be defined in the same way, as they are not equipped with any adjacency structure initially, but equipped with pairwise distances. Moreover, the filtration should be invariant with respect to isometric transformations of point clouds, and we need a different network architecture to that for graphs. While many previous studies, such as Bendich et al. (2007); Cang and Wei (2017); Cang et al. (2018); Meng et al. (2020), have tried to design special filtration in an unsupervised way based on data properties, we aim to learn a filtration in a *data-driven and supervised way*. To the best of our knowledge, this study is the first attempt to learn a filtration for persistent homology on point cloud data in a supervised way.

Another approach to obtaining topological features is to approximate the entire process from (i) to (iii) for computing vectorization of persistent homology by a neural network ( Zhou et al. (2022) and de Surrel et al. (2022)). Note that similar methods are proposed for image data (Som et al., 2020) and graph data (Yan et al., 2022). Since these networks only approximate topological features by classical methods, they cannot extract information more than features by classical methods or extract topological information adaptively to data.

## 2 Background

### 2.1 Network Architectures for Point Clouds

There have been proposed several neural network architectures for dealing with point clouds for their input. Here, we briefly explain DeepSets (Zaheer et al., 2017), which we will use to implement our isometry-invariant structure. A more recent architecture, PointNet (Qi et al., 2017a) and PointMLP (Ma et al., 2022), will also be used for the comparison to our method in the experimental section.

The DeepSets architecture takes a finite (multi)set $X = \{x_1, \ldots, x_N\} \subset \mathbb{R}^d$ of possibly varying size as input. It consists of the composition of two fully-connected neural networks $\phi_1 \colon \mathbb{R}^d \to \mathbb{R}^{d'}$ and $\phi_2 \colon \mathbb{R}^{d'} \to \mathbb{R}^{d''}$ with a permutation invariant operator $\mathbf{op}$ such as $\max, \min$ and summation:

$$\mathrm{DeepSets}(X) \coloneqq \phi_2(\mathbf{op}(\{\phi_1(x_i)\}_{i=1}^N)).$$

For each $x_i \in X$, the map $\phi_1$ gives a representation $\phi_1(x_i)$. These pointwise representations are aggregated via the permutation invariant operator $\mathbf{op}$. Finally, the map $\phi_2$ is applied to provide the network output. The output of DeepSets is permutation invariant thanks to the operator $\mathbf{op}$, from which we can regard the input of the network as a set. The parameters characterizing $\phi_1$ and $\phi_2$ are tuned through the training depending on the objective function.

### 2.2 Persistent Homology

Here, we briefly explain persistence diagrams, weight filtrations for point clouds, and vectorization methods of persistence diagrams.

**Persistence Diagrams** Let $S \colon \mathbb{R}^d \to \mathbb{R}$ be a function on $\mathbb{R}^d$. For $t \in \mathbb{R}$, the $t$-sublevel set of $S$ is defined as $S_t \coloneqq \{p \in \mathbb{R}^d \mid S(p) \leq t\}$. Increasing $t$ from $-\infty$ to $\infty$ gives an increasing sequence of sublevel sets of $\mathbb{R}^d$, called a filtration. *Persistent homology* keeps track of the value of $t$ when topological features, such as connected components, loops, and cavities, appear or vanish in this sequence. For each topological feature $\alpha$, one can find the value $b_\alpha < d_\alpha$ such that the feature $\alpha$ exists in $S_t$ for $b_\alpha \leq t < d_\alpha$. The value $b_\alpha$ (resp. $d_\alpha$) is called the birth (resp. death) time of the topological feature $\alpha$. The collection $(b_\alpha, d_\alpha)$ for topological features $\alpha$ is called the *persistence diagram* (PD) of $f$, which is a multiset of the half-plane $\{(b, d) \mid d > b\}$. The information of connected components, loops, and cavities are stored in the 0th, 1st, and 2nd persistence diagrams, respectively.

Given a point cloud $X \subset \mathbb{R}^d$, i.e., a finite point set, one can take $S$ to be the distance to point cloud, defined by

$$S(z) = \min_{x \in X} \|z - x\|.$$

Then, the sublevel set $S_t$ is equal to the union of balls with radius $t$ centered at points of $X$: $\bigcup_{x \in X} B(x;t)$, where $B(x;t) := \{y \in \mathbb{R}^d \mid \|x - y\| \leq t\}$. In this way, one can extract the topological features of a point cloud $X$ with the persistence diagram. The persistent homology of this filtration can be captured by the filtration called Čech or Vietoris-Rips filtration (Rips filtration for short). In this paper, we use Rips filtration for computational efficiency. See Appendix A for details. The Rips filtration can be computed only by the pairwise distance information.

**Weighted Filtrations for Point Clouds**   While the Rips filtration considers a ball with the same radius for each point, one can consider the setting where the radius value depends on each point (see, for example, Edelsbrunner (1992)). For instance, Hiraoka et al. (2016); Nakamura et al. (2015); Obayashi et al. (2022), which use persistent homology to find the relationship between the topological nature of atomic arrangements in silica glass and whether the atomic arrangement is liquid or solid, use such a filtration. Given a point cloud $X \subset \mathbb{R}^d$, one can define the radius value $r_x(t)$ at time $t$ for $x \in X$ as follows. We first choose a function $w \colon X \to \mathbb{R}$, which is called *weight*, and define

$$r_x(t) := \begin{cases} -\infty & \text{if } t < w(x), \\ t - w(x) & \text{otherwise.} \end{cases}$$

The associated weighted Rips filtration is denoted by $R[X, w]$. One choice as a weight function is the distance-to-measure (DTM) function. The resulting filtration is called a DTM-filtration, for which several theoretical results are shown in Anai et al. (2020). One of those results shows that a persistent homology calculated by the DTM-filtrations is robust to outliers in point clouds, which does not hold for the Rips filtrations.

**Vectorization Methods of Persistence Diagrams**   Persistence diagrams are difficult to handle by machine learning since they are a multiset on a half-plane. For this reason, several vectorization methods of persistence diagrams have been proposed, such as persistence landscape (Bubenik et al., 2015) and persistence image (Adams et al., 2017). Carrière et al. (2020) proposed the method called PersLay, which vectorizes persistence diagrams in a data-driven way using the structure based on DeepSets. In this paper, we utilize this method as a vectorization method. Note that PersLay can be replaced with any vectorization method in our architecture.

Given a function transformation map $\phi_c$ with a learnable parameter $c$ and permutation invariant operator **op**, we can vectorize a persistence diagram $D$ by PersLay with

$$\text{PersLay}(D) := \mathbf{op}(\{\phi_c(q)\}_{q \in D}).$$

If we choose appropriate parametrized function $\phi_c$, we can make PersLay similar to classical vectorization methods such as persistence landscape and persistence image. In this paper, we utilized the following map as $\phi_c$, which makes PersLay similar to persistence image:

$$\phi_c(q) = \left[ \exp\left( -\frac{\|q_1 - c_1\|^2}{2} \right), \ldots, \exp\left( -\frac{\|q_M - c_M\|^2}{2} \right) \right]^\top.$$

## 3   Proposed Framework

In this section, we describe our framework to obtain topological representations of given point clouds. Our idea is to use persistence homology for extracting topological information and model a weight function $w$ in §2.2 by a neural network, which can be learned adaptively to data. Moreover, we propose a way to combine the topological features with deep neural networks.

### 3.1   Filtration Learning for Point Clouds

First, we introduce a structure to learn a filtration on point clouds end-to-end from data. For that purpose, we learn a weight function $w$ for a weighted filtration, by modeling it by a neural network. We assume that the weight function depends on an input point cloud $X = \{x_1, \ldots, x_N\} \subset \mathbb{R}^d$, i.e., $w$ is of the form $f(X, \cdot) \colon \mathbb{R}^d \to \mathbb{R}$. Designing such a function by a neural network has the following three difficulties:

1. The weight function $f(X, \cdot)$ should be determined by the whole point cloud *set X* and does not depend on the order of the points in $X$.

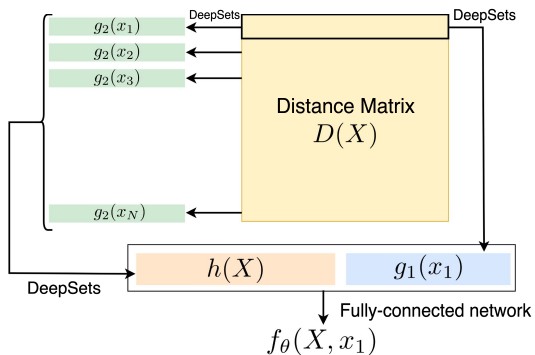

Figure 2: The architecture of the network. This figure is for the case of getting the weight of point $x_1$ in the point cloud $X$. Note that $d(X, x_i)$ is equal to (the transpose of) the $i$-th row of $D(X)$, i.e., $(d(x_i, x_j))_{j=1}^N$.

2. The resulting function should be *isometry-invariant*, i.e, for any isometric transformation $T$, any point cloud $X$, and $x \in X$, it holds that $f(TX, Tx) = f(X, x)$.
3. The output of the $f(X, x)$ should have both of global information $X$ and pointwise information of $x$.

To address the problem 1, we use the DeepSets architecture (§2.1). To satisfy invariance for isometric transformations as in problem 2, we utilize the distance matrix $D(X) = (d(x_i, x_j))_{i,j=1}^N \in \mathbb{R}^{N \times N}$ and the relative distances $d(X, x) = (\|x - x_j\|)_{j=1}^N$ instead of coordinates of points, since $D(TX) = D(X)$ and $d(TX, Tx) = d(X, x)$ hold for any isometric transformation $T$. Finally, to deal with problem 3, we regard $f$ as a function of the feature of $D(X)$ and the feature of $d(X, x)$. Below, we will explain our implementation in detail.

First, we implement a network to obtain a pointwise feature vector of a point $x \in \mathbb{R}^d$ by

$$g_1(x) := \phi^{(2)}(\mathbf{op}(\{\phi^{(1)}(\|x - x_j\|)\}_{j=1}^N)), \tag{1}$$

where $\phi^{(1)}$ and $\phi^{(2)}$ are fully-connected neural networks. This structure (1) can be regarded as the DeepSets architecture whose input is a set of one-dimensional vectors, hence it is permutation invariant.

Next, we use the DeepSets architecture to obtain a feature vector for the entire point cloud $X$ as follows. We first implement pointwise feature vectors with the same architecture as $g_1$, i.e.,

$$g_2(x_i) := \phi^{(4)}(\mathbf{op}(\{\phi^{(3)}(d(x_i, x_j))\}_{j=1}^N)),$$

where $\phi^{(3)}$ and $\phi^{(4)}$ are fully-connected neural networks. Then, we get a feature vector $h(X)$ for the entire point cloud $X$ by applying the DeepSets architecture again:

$$h(X) = \phi^{(5)}(\mathbf{op}(\{g_2(x_i)\}_{i=1}^N)), \tag{2}$$

where $\phi^{(5)}$ is a fully-connected neural network. Here we omit the first inner network for the DeepSets architecture since it can be combined with $\phi^{(4)}$ in $g_2$.

Finally, we implement $f$ as a neural network whose input is the concatenated vector of $h(X)$ and $g_1(x)$. Specifically, we concatenate $h(X)$ with the feature vector $g_1(x)$ for $x \in \mathbb{R}$ to get a single vector, and then input it into a fully-connected neural network $\phi^{(6)}$:

$$f_\theta(X, x) = \phi^{(6)}([h(X), g_1(x)]^\top),$$

where $\theta := (\theta_k)_{k=1}^6$ is the set of parameters that appear in network $\phi^{(k)}$ ($k = 1, \ldots, 6$). The overall neural network architecture is shown in Figure 2.

*Remark* 3.1. Since our implementation of the weight function depends on distance matrices, not on the coordinates of each point, we can apply our method to a dataset given in distance matrix format. In fact, we will apply our framework to such a dataset in Section 5.

*Remark* 3.2. If we want to consider features other than the position information assigned to each point in the calculation of the filtration weights, as in the silica glass studies, we vectorize those features and concatenate them together.

By using the weight function $f_\theta(X, \cdot)$, we can calculate the persistent homology of filtration $R[X, f_\theta(X, \cdot)]$. Then, we can obtain the vectorization using PersLay parametrized by $c$. Then, we can input this vector into a machine learning model $F_{\theta'}$ with parameter $\theta'$. Given the dataset $\{(X_k, y_k)\}_{k=1}^M$, the objective function to be minimized can be written as

$$\mathcal{L}(\theta, \theta', c) := \frac{1}{M} \sum_{k=1}^M \ell\left(F_{\theta'}(\mathrm{PersLay}(R[X_k, f_\theta(X_k, \cdot)])), y_k\right),$$

where $\ell$ is an appropriate loss function depending on tasks. Typically, if the task is regression, $\ell$ is the mean square loss, and if the task is classification, $\ell$ is the cross entropy loss. The optimization problem

$$\min_{\theta, \theta', c} \mathcal{L}(\theta, \theta', c)$$

is solved by stochastic gradient descent. Note that the differentiability of the objective function and the convergence of the optimization algorithm is guaranteed by results in Carrière et al. (2021).

## 3.2 Combining Topological Features with a DNN-based method

While we can use the topological features alone, we can also concatenate them with the features computed by a deep neural network (DNN). In order to do this, we need to learn not only (a) networks in our framework but also (b) DNN. Although (a) and (b) can be learned simultaneously since the output of the resulting feature is differentiable with all parameters, the networks cannot be successfully optimized in our experiments. This would be because the loss function is not smooth with respect to the parameters in (a), which can also make the optimization of (b) unstable. To deal with this issue, we propose to learn (a) and (b) separately.

Let us describe the whole training procedure briefly here. Suppose $X$ is a input point cloud. We write $\Psi_{\mathrm{topo}}(X) \in \mathbb{R}^{L_1}$ for the topological feature which comes from our method, and $\Psi_{\mathrm{DNN}}(X) \in \mathbb{R}^{L_2}$ for the feature that comes from a DNN-based method. Let $\ell$ be a loss function for the classification task and $n$ be a number of classes. We propose the following two-phase training procedure:

**1st Phase.** Let $C_1 \colon \mathbb{R}^{L_2} \to \mathbb{R}^n$ be a classifier that receives feature from $\Psi_{\mathrm{DNN}}$. Then, $C_1$ and $\Psi_{\mathrm{DNN}}$ are learned through the classification task, which is achieved by minimizing $\sum_{k=1}^M \ell(C_1(\Psi_{\mathrm{DNN}}(X_k)), y_k)$.

**2nd Phase.** Let $C_2 \colon \mathbb{R}^{L_1+L_2} \to \mathbb{R}^n$ be a classifier. *We discard the classifier $C_1$ and fix $\Psi_{\mathrm{DNN}}$.* Then, $C_2$ and $\Psi_{\mathrm{topo}}$ are learned through the classification task, which is achieved by minimizing $\sum_{k=1}^M \ell(C_2([\Psi_{\mathrm{DNN}}(X_k)^\top, \Psi_{\mathrm{topo}}(X_k)^\top]^\top), y_k)$.

This training procedure is observed to make the optimization stable and to help the architecture to achieve higher accuracy. We show the experimental result in § 5.

## 4 Approximation Ability of Weight Function

In this section, we theoretically show that our architecture has a good expression power. We prove that approximation of any continuous map can be realized by our idea to concatenate two finite-dimensional feature vectors.

Consider the space $2^{[0,1]^m}$ of subsets in $[0,1]^m$ equipped with the Hausdorff distance. For a fixed $N \in \mathbb{N}$, we define the following subspace of $2^{[0,1]^m}$:

$$\mathcal{X} := \{X \subset [0,1]^m \mid |X| \leq N\}.$$

The following theorem states that we can approximate any continuous function on $\mathcal{X} \times [0,1]^m$ by concatenating the finite dimensional feature vectors of a point cloud and a point. The proof of the theorem is given in Appendix B.

**Theorem 4.1.** *Let $f\colon \mathcal{X} \times [0,1]^m \to \mathbb{R}$ be a continuous function. Then for any $\epsilon > 0$, there exist $K \in \mathbb{N}$ and continuous maps $\psi_1\colon \mathcal{X} \to \mathbb{R}^K, \psi_2\colon \mathbb{R}^{K+m} \to \mathbb{R}$ such that*

$$\sup_{X \in \mathcal{X}} \int_{[0,1]^m} \left( f(X,x) - \psi_2([\psi_1(X)^\top, x^\top]^\top) \right)^2 \mathrm{d}x < \epsilon.$$

## 5 Experiments

To investigate the performance of our method in classification tasks, we conducted numerical experiments on two types of public real-world datasets. In §5.1, we present a simple experiment on a protein dataset to demonstrate the efficacy of our method. Next, in §5.2, we show the result of a experiment on a 3D CAD dataset. The classification accuracy was used as the evaluation metric.

The first experiment was conducted to show the validity of our method where we only used the topological feature, without DNN features. In the second experiment, on the other hand, we combine the topological features and features from DNNs using the method described in §3.2.

All the code was implemented in Python 3.9.13 with PyTorch 1.10.2. The experiments were conducted on CentOS 8.1 with AMD EPYC 7713 2.0 GHz CPU and 512 GB memory. Persistent homology was calculated by Python interface of GUDHI[1]. All of the source codes we used for experiments is publically available [2]. More details of the experiments can be found in Appendix C.

### 5.1 Protein classification

**Dataset.** In this section, we utilize the protein dataset in Kovacev-Nikolic et al. (2016) consisting of dense configuration data of 14 types of proteins. The authors of the paper analyzed the maltose-binding protein (MBP), whose topological structure is important for investigating its biological role. They constructed a dynamic model of 370 essential points each for these 14 types of proteins and calculated the cross-correlation matrix $C$ for each type[3]. They then define the distance matrix, called the *dynamical distance*, by $D_{ij} = 1 - |C_{ij}|$, and they use them to classify the proteins into two classes, "open" and "close" based on their topological structure. In this paper, we subsampled 60 points and used the distance matrices with the shape $60 \times 60$ for each instance. We subsampled 1,000 times, half of which were from open and the rest were from closed. The off-diagonal elements of the distance matrix were perturbed by adding noise from a normal distribution with a standard deviation of 0.1.

**Architectures and comparison baselines.** Given a distance matrix, we computed the proposed adaptive filtration and calculate the persistent homology. Then, the 1st persistence diagram of that filtration was vectorized by PersLay. The obtained feature was input into a classifier using a linear model, and all the models were trained with cross entropy loss. For comparison, we evaluate the accuracy when we replace our learnable filtration with a fixed filtration. As a fixed filtration, we used Rips or DTM filtration (Anai et al., 2020), the latter of which is robust to outliers. The hyperparameter for DTM filtration was set to maximize the classification accuracy. Additionally, we replaced our topological feature by the output of (2), which we call DistMatrixNet, and compared the classification accuracy. Note that we used DistMatrixNet not for the computation of filtration weights but for the direct computation of point clouds' feature. Since the protein dataset is given in the format of distance matrices, we did not utilize standard neural network methods for point clouds.

**Results.** We present the results of the binary classification task of proteins in Table 1. We can see that our framework overperformed the other methods in terms of the accuracy of the classification.

We describe some observations below: Firstly, the result of our method is better than those of Rips and DTM filtrations. This would be because our method learned a weighted filtration adaptively to data and tasks and could provide more informative topological features than the classical ones. Secondly, our method achieved higher accuracy compared to DistMatrixNet, which has a similar

---

[1] https://gudhi.inria.fr/

[2] https://github.com/git-westriver/FiltrationLearningForPointClouds

[3] The correlation matrix $C$ can be found in https://www.researchgate.net/publication/301543862_corr.

Table 1: The accuracy of the binary classification task of protein structure. We compared our methods with DistMatrixNet and persistent homology with Rips/DTM filtration. We can see that our method performs better than other three methods.

| DistMatrixNet | Rips | DTM | Ours |
|---|---|---|---|
| 65.0 ± 12.0 | 79.9 ± 3.0 | 78.0 ± 1.6 | **81.9 ± 2.1** |

number of parameters to the proposed method. This suggests that persistent homology is essential in the proposed method rather than the expressive power of DistMatrixNet.

## 5.2 3D CAD data classification

**Dataset.** In this section, we deal with the classification task on ModelNet10 (Wu et al., 2015). This is a dataset consisting of 3D-CAD data given by collections of polygons of 10 different kinds of objects, such as beds, chairs, and desks. We choose 1,000 point clouds from this dataset so that the resulting dataset includes 100 point clouds for each class. The corresponding point cloud was created with the sampling from each polygon with probability proportional to its area. Moreover, to evaluate the performance in the case where the number of points in each point cloud is small, we subsampled 128 points from each point cloud. The number of points is relatively small compared with the previous studies, but this setting would be natural from the viewpoint of practical applications. We added noise to the coordinates of each sampled point using a normal distribution with a standard deviation of 0.1.

**Architecture and comparison baselines.** We utilized the two-phase training procedure described in §3.2, where we concatenated a feature computed by DNN-based method and a topological feature. As DNN-based methods, which were trained through the 1st Phase, we utilized DeepSets (Zaheer et al., 2017), PointNet (Qi et al., 2017a), and PointMLP (Ma et al., 2022). A given point cloud was input into our adaptive filtration architecture, and the resulting 1st persistence diagram was vectorized by PersLay. The concatenated features were input into a classifier based on a linear model, and the parameters are tuned with the cross entropy loss.

We compared the result of 1st Phase, which is the accuracy of a DNN alone, and that of 2nd Phase to validate the efficacy of our learnable filtration. We also compared our method with a fixed filtration, Rips or DTM, instead of our learnable filtration. Furthermore, we replaced our topological feature replaced by the output of DistMatrixNet (2) and compared the classification accuracy.

**Results.** The results are shown in Table 2. Below, we present some observations for the results.

Firstly, our method outperformed Rips filtration, which is one of the most common filtrations. Moreover, the accuracy of our method is almost the same as or higher than that of DTM filtration. While DTM filtration has hyperparameters that need to be tuned, our method achieved such a result by learning an adaptive filtration automatically through the training. This result implies that our method is useful in efficiently choosing filtrations.

Secondly, our method again overperformed DistMatrixNet. This indicates that the combination of persistent homology and DistMatrixNet was crucial similarly to the protein dataset.

Thirdly, and most importantly, it can be observed that the classification accuracy is better when our method was concatenated with DeepSets/PointNet compared to using DeepSets/PointNet alone. Additionally, the accuracy when we combine our method and DeepSets/PointNet is competitive when using PointMLP alone. These observations indicate that concatenating the topological features obtained by our method yields positive effects. The accuracy when we combine our method and PointMLP is not higher than that of PointMLP. This would mean that PointMLP has already captured enough information including topological features during the 1st phase, so that the information obtained by persistent homology may be redundant, potentially negatively impacting the classification.

## 6 Conclusion

In this paper, we tackled the problem to obtain adaptive topological features of point clouds. To this end, we utilize a weighted filtration and train a neural network that generates a weight for each point.

Table 2: Results for the classification task of 3D CAD data. The row named "1st Phase" shows the results when we used the feature calculated by DNN only, and the rows named "2nd Phase" shows the results when we concatenated them with the feature calculated by DistMatrixNet or topological feature obtained by persistent homology. Compared to using the DeepSets/PointNet alone, we can achieve higher accuracy when we concatenate the topological feature obtained by our method. Additionally, regardless which DNN method we use, using topological feature computed by our method improves accuracy better than using topological feature computed by Rips filtration and feature computed by DistMatrixNet. Moreover, the accuracy when we combine our method with DeepSets/PointNet is competitive with using PointMLP alone.

|  |  | DeepSets | PointNet | PointMLP |
|---|---|---|---|---|
| # of parameters |  | 813,488 | 3,472,500 | 3,524,386 |
| 1st Phase |  | 65.7 ± 1.4 | 64.3 ± 4.4 | **68.8 ± 6.3** |
| 2nd Phase | DistMatrixNet | 65.7 ± 4.8 | 55.7 ± 13.9 | 53.8 ± 7.4 |
|  | Rips | 67.0 ± 2.6 | 68.4 ± 2.4 | 57.8 ± 12.4 |
|  | DTM | **68.0 ± 2.5** | 68.7 ± 2.3 | 57.2 ± 6.8 |
|  | Ours | **67.5 ± 2.5** | **68.8 ± 2.0** | 60.0 ± 6.3 |

Since the model is desired to be invariant with respect to isometric transformation, we proposed a network architecture that satisfies such invariance. We theoretically showed a finite-dimensional approximation property for a wide class of weight functions, which supports the validity of our architecture. Our numerical experiments demonstrated the effectiveness of our method in comparison with the existing methods.

**Limitations and future work** There are several limitations and remained future work for this study. First, while we concentrated on determining the weight function of the weighted Rips filtration in this paper, we can also consider other kinds of filtrations. For example, we can expand the balls from each point at different speeds. Second, we did not applied our method to larger datasets as we needed to compute persistent homology repeatedly in the training, which make the computational cost higher. In fact, it took about seven hours to train the neural networks in our method for each cross-validation. While we believe our method is meaningful since it sometimes improves classification accuracy, this is one of the largest limitations of our study. Addressing this issue is one of our future work. Third, since our method adaptively chooses a filtration, our framework is expected to work well even if a subsequent machine learning model has a simple architecture. We plan to validate the hypothesis experimentally and theoretically.

## Acknowledgments

We appreciate Marc Glisse and Théo Lacombe for the helpful discussion. This work was partially supported by Grant-in-Aid for Transformative Research Areas (A) (22H05107) and JST KAKENHI JP19H01114.

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
