# OpenReview forum: "Adaptive Topological Feature via Persistent Homology: Filtration Learning for Point Clouds"
_NeurIPS.cc/2023/Conference — NeurIPS 2023 poster_

### Official Review · Reviewer_q2Rp · 2023-06-23

**Soundness:** 3 good
**Presentation:** 3 good
**Contribution:** 3 good
**Rating:** 7
**Confidence:** 4

**Summary:**

In this manuscript, the authors develop a special module that can adaptively learn a suitable filtration function for persistent homology (PH) and its downstream tasks. In particular, their learning module is specially designed, so that resulting persistent homology is isometry invariant. Their model has been tested on two datasets for protein classification and CAD data classification. The model is novel and very interesting!

**Strengths:**

It is a novel approach to use machine learning to learn a suitable filtration function! It has also demonstrated the advantage over traditional approaches.

**Weaknesses:**

Missing important references for related works. The test examples have limited data points. The model may suffer from over-fitting issues.

**Questions:**

1.	One of the key point for the submission is to "propose a novel framework to obtain adaptive topological features for point clouds", as mentioned in Page 3, line 71. However, the submission seems to only focus on learning "weight function" using "different radii"! In fact, many other works have been done for designing special filtration process based on data properties, such as local homology, weighted homology, element-specific homology, cohomology (incorporated with special weights), etc. These works can all be viewed as learning (unsupervisedly) "adpative topological features".  For instance,

(a) local homology:  Bendich P, Cohen-Steiner D, Edelsbrunner H, Harer J, Morozov D (2007) Inferring local homology from sampled stratifed spaces. In foundations of computer science, 2007. FOCS’07. 48th Annual IEEE symposium on, IEEE, pp. 536–546

(b) element-specific homology & special distance matrices: Cang ZX, Mu L, Wei GW (2018) Representability of algebraic topology for biomolecules in machine learning based scoring and virtual screening. PLoS Comput Biol 14:e1005929

Cang ZX, Wei GW (2017) TopologyNet: Topology based deep convolutional and multi-task neural networks for biomolecular property predictions. PLoS Comput Biol 13:e1005690

(c) Weighted simplicial complexes: Zhenyu Meng, D Vijay Anand, Yunpeng Lu, Jie Wu, and Kelin Xia, "Weighted persistent homology for biomolecular data analysis." Scientific Report, 10 (1), 1-15 (2020)

(d) Topological antoencoders: Moor, Michael, Max Horn, Bastian Rieck, and Karsten Borgwardt. "Topological autoencoders." In International conference on machine learning, pp. 7045-7054. PMLR, 2020.

Further, "adaptive topological features for point clouds" can also been achieved through different types of simplicial complexes or hypergraphs. For instance, the topological features of Rips complexes are dramatically different from Dowker complexes, Neighborhood complexes, Hom-complexes, etc.

Xiang Liu, Huitao Feng, Jie Wu, and Kelin Xia, "Dowker complex based machine learning (DCML) models for protein-ligand binding affinity prediction." PLOS Computational Biology, 18(4), e1009943

Xiang Liu, and Kelin Xia, "Neighborhood complex based machine learning (NCML) models for drug design." In Interpretability of Machine Intelligence in Medical Image Computing, and Topological Data Analysis and Its Applications for Medical Data, pp. 87-97. Springer, Cham (2021).

Xiang Liu, Huitao Feng, Jie Wu, and Kelin Xia, "Hom-complex-based machine learning (HCML) for the prediction of protein–protein binding affinity changes upon mutation", Journal of Chemical Information and Modeling, 62 (17), 3961-3969 (2022)

2.	The discussions of vectorization of PH are not proper. In particular, “The idea to learn vectorization of persistent homology was pioneered by Hofer et al. (2017” is incorrect. To learn statistic or combinatorial properties (unsupervised) from persistent diagram or persistent barcodes is a common approach. For instance,

Bubenik, Peter, and Peter T. Kim. "A statistical approach to persistent homology." Homology, homotopy and Applications 9, no. 2 (2007): 337-362.

Chung, Moo K., Peter Bubenik, and Peter T. Kim. "Persistence diagrams of cortical surface data." In Information Processing in Medical Imaging: 21st International Conference, IPMI 2009, Williamsburg, VA, USA, July 5-10, 2009. Proceedings 21, pp. 386-397. Springer Berlin Heidelberg, 2009.

Bubenik P (2015) Statistical topological data analysis using persistence landscapes. J Mach Learn Res 16:77–102
Chi Seng Pun, Si Xian Lee, and Kelin Xia, "Persistent-homology-based machine learning: a survey and a comparative study." Artificial Intelligence Review, (2022)

Dey, Tamal Krishna, and Yusu Wang. Computational topology for data analysis. Cambridge University Press, 2022.

3.	In the test examples, the data sizes seem to be relatively small, i.e., less than 2000 datapoints. But the learning module has multiple fully connected layers. In this way, overfitting can easily be a problem. The authors are suggested to do some ablation studies to address the issue. Further, some more details for PH analysis are needed. For instance, filtration sizes for the data, vectorization of persistent images, etc.

**Limitations:**

It will be great if the authors can use more realistic examples and compare with state-of-the-art models.

---

> ### Author Rebuttal · Authors · 2023-08-09
>
> We appreciate the detailed comments and suggestions for improving our paper. We will reflect all the comments and suggestions in our final version. In the following, we respond to specific concerns and questions raised by the reviewer.
>
> > 1. One of the key point for the submission is to "propose a novel framework to obtain adaptive topological features for point clouds", as mentioned in Page 3, line 71. However, the submission seems to only focus on learning "weight function" using "different radii"! In fact, many other works have been done for designing special filtration process based on data properties, such as local homology, weighted homology, element-specific homology, cohomology (incorporated with special weights), etc. These works can all be viewed as learning (unsupervisedly) "adpative topological features".
>
> Thank you for pointing out that the expression “adaptive topological feature” might be confusing for readers, because "adaptive topological features" does not necessarily mean that the extraction of topological features is done in a supervised way.
>
> In this paper, we tried to learn a filtration in a data-driven and supervised way, which is our main contribution. This is not completely on the same line with previous studies designing special filtration in an unsupervised way based on data properties. We missed some of the important references you cited and will add them in the final version.
>
> > Further, "adaptive topological features for point clouds" can also been achieved through different types of simplicial complexes or hypergraphs. For instance, the topological features of Rips complexes are dramatically different from Dowker complexes, Neighborhood complexes, Hom-complexes, etc.
>
> While we concentrated on determining the weight function of the weighted Rips filtration in this paper, we can also consider other kinds of filtrations as you pointed out. We will clearly state this remark in the limitation section in the final version.
>
> > 2. The discussions of vectorization of PH are not proper. In particular, “The idea to learn vectorization of persistent homology was pioneered by Hofer et al. (2017” is incorrect. To learn statistic or combinatorial properties (unsupervised) from persistent diagram or persistent barcodes is a common approach.
>
> While we intended to state that Hofer et al. (2017) first proposed to determine vectorization in an end-to-end/data-driven/supervised/automatic way, the sentence you pointed out is inappropriate. In the final version, we will modify this expression and add references related to the vectorization method.
>
> > 3. In the test examples, the data sizes seem to be relatively small, i.e., less than 2000 datapoints. But the learning module has multiple fully connected layers. In this way, overfitting can easily be a problem. The authors are suggested to do some ablation studies to address the issue.
>
> Thank you for your important suggestion. We will conduct additional experiments to investigate whether we can reduce the number of network parameters, and add the results in the appendix.
>
> > Further, some more details for PH analysis are needed. For instance, filtration sizes for the data, vectorization of persistent images, etc.
>
> Since we used (weighted) Rips filtration, the number of simplices can be computed from the number of points in the point clouds, which is stated in lines 273 and 302. To make it clear, we will add this in the appendix in the final version.
>
> We used PersLay (Carriere et al., 2019) as a vectorization method, and the detailed settings for this are described in Appendix B.3.
>
>
> We hope that we addressed all your questions and concerns adequately. In light of our clarifications, please consider increasing your score to accept. Please let us know if we can provide any further details and/or clarifications.

---

> > ### Comment · Reviewer_q2Rp · 2023-08-11
> >
> > Thanks for the reply! I have no further comments.

---

### Official Review · Reviewer_k7w9 · 2023-07-06

**Soundness:** 3 good
**Presentation:** 2 fair
**Contribution:** 2 fair
**Rating:** 4
**Confidence:** 4

**Summary:**

The paper develops a neural network to learn weights of given points in addition to other internal parameters to classify 3D clouds of unlabeled points on several public datasets.


**Strengths:**

The authors should be highly praised for a rigorous approach to an important problem of point cloud classification by using isometry invariants from persistent homology.

The paper is generally well-written.

**Weaknesses:**

Questions arise already when reading the abstract in line 11: "to make the resulting persistent homology isometry-invariant, we develop a neural network". For standard filtrations such as Vietoris-Rips, Cech, or Delaunay complexes on a point cloud, the persistent homology is already an isometry invariant of a given cloud of unlabeled points because constructions of all complexes above depend only on inter-point distances.

The main drawback of persistent homology is its weakness as an isometry invariant, which should have been clear to all experts in computational geometry many years ago but was demonstrated only recently. The paper by Smith et al (arxiv:2202.00577) explicitly constructs generic families of point clouds in Euclidean and metric spaces that are indistinguishable by persistence and even have empty persistence in dimension 1.

Though Topological Data Analysis was largely developed by mathematicians, the huge effort over many years was invested into speeding up computations, rather surprisingly, instead of trying to understand the strengths and weaknesses of persistent homology, especially in comparison with the much simpler, faster, and stronger invariants of clouds under isometry.

Persistence in dimension 0 was actually extended to a strictly stronger invariant mergegram by Elkin et al in MFCS 2020 and Mathematics 2021, which has the same asymptotic time as the classical 0D persistence and is also stable under perturbations of points.

A SoCG 2022 workshop included a frank discussion concluding that there was no high-level problem that persistent homology solves. In fact, persistence as an isometry invariant essentially tries to distinguish clouds up to isometry, not up to continuous deformations since even non-uniform scaling changes persistence in a non-controllable way.

On the other hand, the isometry classification problem for point clouds was nearly solved by Boutin and Kemper (2004), who proved that the total distribution of pairwise distances is a generically complete invariant of point clouds in any Euclidean space. The remaining singular cases were recently covered by Widdowson et al in NeurIPS 2022 and CVPR 2023.

**Questions:**

Are there any theoretical guarantees that the output of the proposed neural network distinguishes infinitely many (almost all?) point clouds as pairwise distances do?

What theoretical results in the paper are stronger than the past work by Widdowson et al in CVPR 2023?

In subsection 5.1, what is the dataset size relative to the Protein Data Bank?

In subsection 5.2, how representatives are "subsampled 128 points" (line 302) for objects such as "beds, chairs, and desks" (line 298). For example, can humans distinguish between a bed and a desk by looking at 128 randomly sampled points?

What is the asymptotic complexity (in the number of points in a given cloud) of the algorithm described in section 3? What was the actual running time for training and testing, and what are the technical specifications of the used machine?

**Limitations:**

The very last paragraph on limitations only discusses talks about future work, for example about "generalizing our framework to include a wider class of weighted filtrations" (line 342). Yes, there are numerous papers that go even further and include a given point cloud as a raw input without computing any justified invariants.

Since the authors have written detailed and accurate mathematical proofs in the appendix, they could probably agree that *examples prove nothing* because counter-examples can still exist, especially when all possible data (as for point clouds) fill a continuous space.

For continuous data, a continuous parametrization or metric can be more suitable than a discrete classification, which practically cuts a continuous space into disjoint pieces.

Since all tables of experimental results include accuracies of at most 84% in table 1 (maximum 75% in table 2), the best and certainly publishable contribution seems to be Theorem 4.1. Can a mathematical venue be more suitable for this important result?

To help the authors with future submissions, the key insight from "AlphaFold2 one year on" https://www.nature.com/articles/s41592-021-01365-3 exposes the major limitation of all brute-force predictions, not only for AlphaFold. What resources (money, people, even electricity and water) are needed not only to get the first predictions but to annually update predictions with new training data? Is it really sustainable in the long term?

---

> ### Author Rebuttal · Authors · 2023-08-09
>
> We are grateful for the detailed comments and suggestions for improving our paper. First, let us explain the problem setting and our motivation for this study to resolve your misunderstanding.
>
> In this paper, we deal with the classification of *labeled point clouds.* In this setting, *point clouds are labeled beforehand* (for example, in the experiment on protein dataset, each point cloud has a label open or close), and two point clouds with the same label are not necessarily isometric. Our aim is NOT to distinguish the point clouds up to isometry.
>
> In solving such classification tasks, topological features extracted by persistent homology (PH) would be useful. In fact, PH has been shown to be *effective for point cloud analysis and classification* for material science [1, 2], biology [3, 4], and medical science [5, 6]. Although it is isometry-invariant if the filtration is chosen to be isometry-invariant, the use of PH is NOT limited to distinguishing the point clouds up to isometry.
>
> [1] T. Nakamura et al. Persistent homology and many-body atomic structure for medium-range order in the glass. *Nanotechnology*, 26(30):304001, 2015.
>
> [2] A. Hirata et al. Structural changes during glass formation extracted by computational homology with machine learning. *Communications Materials*, 1(1):98, 2020.
>
> [3] V. Kovacev-Nikolic et al. Using persistent homology and dynamical distances to analyze protein binding. *Statistical Applications in Genetics and Molecular Biology,* 15(1):19–38, 2016.
>
> [4] Z. Cang and G. Wei. "TopologyNet: Topology based deep convolutional and multi-task neural networks for biomolecular property predictions." *PLoS computational biology* 13(7), e1005690, 2017
>
> [5] X. Zhu et al. Stochastic Multiresolution Persistent Homology Kernel. In *IJCAI 2016* (pp. 2449-2457).
>
> [6] N. Singh et al. Topological descriptors of histology images. In *Machine Learning in Medical Imaging: 5th International Workshop, MLMI 2014. Proceedings 5* (pp. 231-239).
>
> > the persistent homology is already an isometry invariant
>
> We consider weighted Rips filtration whose weight function is implemented by the neural network. If we do not force this network to be isometry-invariant, the resulting PH can be non-isometry-invariant. This is why we make the network to be isometry-invariant.
>
> > What theoretical results in the paper are stronger than the past work by Widdowson et al … ?
>
> The result by Widdowson et al (2023) is not a competitor of our study but rather one that could potentially improve our results by incorporating it into our study. Currently, we do not have strict guarantees about the approximation ability of our network based on distance matrices. Widdowson et al (2023) showed that Simplexwise Centered Distribution (SCD) is complete isometry invariant and continuous. This helps us to obtain theoretical guarantees of our network or to propose a new network architecture with stronger theoretical guarantees. That is our future work, and we are grateful for your insight. We will mention this in the final version.
>
> > In subsection 5.1, what is the dataset size relative to the Protein Data Bank?
>
> As explained in lines 266—276, we created the protein dataset composed of 1,000 data by subsampling from 14 types of proteins. This is much smaller than the datasets in Protein Data Bank. We did not applied our method to larger datasets since we need to compute PH repeatedly in the training, which makes the computational cost higher. We will add this limitation in the final version.
>
> > In subsection 5.2, how representatives are "subsampled 128 points" …, can humans distinguish between a bed and a desk … ?
>
> Thank you for your important comment. Subsampling 128 points sometimes makes it hard for humans to judge the label for each point cloud. We subsampled a relatively small number of points due to the high computational cost, as we described above. It is our future work to conduct experiments for point clouds with a large number of points.
>
> > What is the asymptotic complexity … ?
>
> The computational complexity can be upper-bounded by $O(N^9)$, where $N$ is the number of points. However, actual computational complexity is expected to be smaller since computational cost of PH is known to be empirically less than cubic complexity with respect to the number of simplices.
>
> Regarding the actual running time, it takes about seven hours to train the neural networks in our method. The bottleneck of computational time would be the computation of PH. While this is one of the limitations of our study, we believe our method is meaningful since it sometimes improves classification accuracy.
>
> We will include this information related to the computational cost in the final version.
>
> The technical specifications of the used machine are described in lines 261—264.
>
> > The very last paragraph on limitations only discusses talks about future work
>
> Thank you for pointing out the lack of a description in the limitation. We will add some explanation about the limitation of our method such as the high computational cost.
>
> > Are there any theoretical guarantees …
>
> > Since the authors have written …
>
> Our method aims to estimate labels for meaningful labeled point clouds, rather than distinguishing between all possible point configurations up to isometry. Our theory supports the validity of the architecture to solve such a classification task. We believe our method is effective for this task.
>
> > Since all tables …
>
> Due to the high computational cost of our method, we could not apply our method to large point clouds, which makes the accuracy lower. If we improve our method so that we can apply it to larger point clouds, the accuracy will be much higher. We will state this fact in the limitation section of the final version.
>
> We hope that we addressed your questions and concerns. In light of our clarifications, please consider increasing your score to accept. Please let us know if we can provide any further details and/or clarifications.

---

> > ### Comment · Reviewer_k7w9 · 2023-08-12
> > **further questions**
> >
> > Thank you for the reply.
> >
> > >Our aim is NOT to distinguish the point clouds up to isometry.
> >
> > Have you checked that all your input clouds from different classes are distinguished by persistent homology? If not, all further outputs for these indistinguishable clouds will be identical.
> >
> > > topological features extracted by persistent homology (PH) would be useful
> >
> > How can these features be called topological if PH changes even under non-uniform scaling, much worse under more flexible topological transformations?
> >
> > >PH has been shown to be effective for point cloud analysis and classification for material science [1, 2], biology [3, 4], and medical science [5, 6].
> >
> > Could you please give exact references to rigorously proved theorems in the cited papers that show the effectiveness of PH?
> >
> > >Although it is isometry-invariant if the filtration is chosen to be isometry-invariant, the use of PH is NOT limited to distinguishing the point clouds up to isometry.
> >
> > If PH is not limited to distinguishing the point clouds up to isometry, under what other equivalence relation is PH an invariant?
> >
> > > If we do not force this network to be isometry-invariant, the resulting PH can be non-isometry-invariant.
> >
> > If the output is not an isometry invariant, does it have other theoretical guarantees?
> >
> > >This is why we make the network to be isometry-invariant.
> >
> > Have you compared the output with the much simpler and faster isometry invariants such as the total distribution of pairwise distances, which was proved to be complete for Euclidean clouds in general position by Boutine and Kemper in Adv. Appl. Math (2004)?
> >
> > > we created the protein dataset composed of 1,000 data by subsampling from 14 types of proteins. This is much smaller than the datasets in Protein Data Bank.
> >
> > Yes, the PDB started in 1971 with 7 structures and now has more than 200 thousands.
> >
> > >The computational complexity can be upper-bounded by O(N^9), where is the number of points.
> >
> > Where is this claim proved in the submission?
> >
> > Is it possible to explain the conflicting quotes from the rebuttal below and justify any claimed beliefs by rigorous arguments?
> >
> > Quote 1. "Due to the high computational cost of our method, we could not apply our method to large point clouds, which makes the accuracy lower."
> >
> > Quote 2. "We believe our method is effective for this task."
> >
> > Quote 3. "We did not claim that we had proposed a new and efficient method to analyze protein datasets."

---

> > > ### Author Response · Authors · 2023-08-15
> > >
> > > Thank you for your comments.
> > >
> > > > Have you checked that all your input clouds from different classes are distinguished by persistent homology? If not, all further outputs for these indistinguishable clouds will be identical.
> > >
> > > There is no need to completely classify point clouds using only PH. If the classification based solely on PH information is insufficient, it can be combined with other features, such as features from DNN. In this case, PH is not used to perform classification on its own, but rather plays a supportive role in the classification. We have discussed this in Section 3.2 of our paper.
> > >
> > > > Could you please give exact references to rigorously proved theorems in the cited papers that show the effectiveness of PH?
> > >
> > > In our paper, we address the machine learning problem, and we do not aim to solve any mathematical problem. Our study is based on previous research showing the empirical effectiveness of PH.
> > >
> > > > If PH is not limited to distinguishing the point clouds up to isometry, under what other equivalence relation is PH an invariant?
> > >
> > > The fact that PH is isometry-invariant does not mean it can only be used to distinguish point clouds up to isometry. In our study, we attempt to use features obtained from PH for classification. It is not directly relevant to our study whether there are any other equivalence relations for which PH is invariant.
> > >
> > > > If the output is not an isometry invariant, does it have other theoretical guarantees?
> > >
> > > Our output is isometry-invariant.
> > >
> > > > Have you compared the output with the much simpler and faster isometry invariants such as the total distribution of pairwise distances, which was proved to be complete for Euclidean clouds in general position by Boutine and Kemper in Adv. Appl. Math (2004)?
> > >
> > > As well as the result by Widdowson et al. (2023), the work of Boutine and Kemper (2004) is not a competitor of our study but rather one that could potentially improve our results by incorporating it into our study. We will also mention this in the final version.
> > >
> > > > Where is this claim proved in the submission?
> > >
> > > This would be well-known in the field of computational topology. In the rebuttal, we wrote, “We will include this information related to the computational cost in the final version.”
> > >
> > > > Is it possible to explain the conflicting quotes from the rebuttal below and justify any claimed beliefs by rigorous arguments?
> > > >
> > > > Quote 1. "Due to the high computational cost of our method, we could not apply our method to large point clouds, which makes the accuracy lower."
> > > >
> > > > Quote 2. "We believe our method is effective for this task."
> > > >
> > > > Quote 3. "We did not claim that we had proposed a new and efficient method to analyze protein datasets."
> > >
> > > The three quotes you mentioned are not contradictory. We provide further details as follows.
> > >
> > > Quote 1: We proposed the idea of learning filtration to achieve higher classification accuracy. We demonstrated its effectiveness for datasets with a small number of points. However, applying this to large datasets is currently challenging due to computational costs. This is our future work.
> > >
> > > Quote 2: We believe that learning filtration makes it possible to extract more suitable information for classification by PH, which leads to an improvement in classification accuracy. We indeed demonstrated its validity in our experiments.
> > >
> > > Quote 3: The experiments on the protein dataset were presented as an example to demonstrate the effectiveness of learning filtration. We do not aim to propose an "efficient" method from a resource consumption perspective. This quote is meant to address your review query regarding the sustainability of our research in terms of resources.

---

> > > > ### Comment · Reviewer_k7w9 · 2023-08-18
> > > > **further questions**
> > > >
> > > > Thank you for the detailed reply.
> > > >
> > > > >we deal with the classification of labeled point clouds. In this setting, point clouds are labeled beforehand (for example, in the experiment on protein dataset, each point cloud has a label open or close)
> > > >
> > > > Is this classification subjective, not based on a rigorously defined equivalence?
> > > >
> > > > >If the classification based solely on PH information is insufficient, it can be combined with other features, such as features from DNN. In this case, PH is not used to perform classification on its own, but rather plays a supportive role in the classification
> > > >
> > > > Have you checked that that the combined features distinguish all point clouds in your data?
> > > >
> > > > >we do not aim to solve any mathematical problem.
> > > >
> > > > What problem did the paper aim to solve?
> > > >
> > > > >The fact that PH is isometry-invariant does not mean it can only be used to distinguish point clouds up to isometry.
> > > >
> > > > How can PH be used for another equivalence if PH is not invariant under this equivalence?
> > > >
> > > > >Our output is isometry-invariant.
> > > >
> > > > Then does it makes sense to compare your output with the past work on isometry invariants? Table 2 includes comparisons with the past work on Rips, DTM, DistMatrixNet. Is the output of DistMatrixNet invariant under permutation of points?
> > > >
> > > > Are there any guarantees that Rips and DTM don't map non-equivalent inputs to the same output? Such guarantees where proved for the distribution of pairwise distances on almost all Euclidean clouds by Boutin and Kemper and Adv. Appl. Math. (2004).
> > > >
> > > > >the work of Boutin and Kemper (2004) is not a competitor of our study
> > > >
> > > > This past work seems much stronger because the distribution of all pairwise distances needs only O(m^2) time on m unordered points, is Lipschitz continuous under perturbations of given and generically complete as proved nearly 20 years ago. Did you know about this work?
> > > >
> > > > >The computational complexity can be upper-bounded by  O(N^9), where N is the number of points. This would be well-known in the field of computational topology.
> > > >
> > > > If this result is well-known, is it possible to give a reference?
> > > >
> > > > >We proposed the idea of learning filtration to achieve higher classification accuracy.
> > > >
> > > > Does higher accuracy on a finite dataset guarantee a similarly high accuracy on other or larger datasets?
> > > >
> > > > >We did not really understand the sentence “Since the authors have written detailed and accurate mathematical proofs in the appendix, they could probably agree that examples prove nothing because counter-examples can still exist, especially when all possible data (as for point clouds) fill a continuous space.” in Limitations. Although we tried to answer this within our understanding, but could you explain it in more detail?
> > > >
> > > > Here is a practical illustration. The prediction that any odd integer (greater than 2) is prime (has no divisors greater than 1) works well for 3,5,7, not 9 (an experimental error), 11, 13, hence achieving the accuracy of 5/6, more than 80%. One can easily improve the accuracy by forbidding multiples of 3 (greater than 3). Then the first counter-example will be 35. One can easily continue forbidding multiples of 5,7, and so on, making the accuracy nearly 100%. Should we use such an algorithm for detecting primes in cryptography-based financial transactions? No because the infinite set of primes will always provide counter-examples. The modern world exists due to mathematically proved results, not due to unjustified predictions. Examples prove nothing because (infinitely many) counter-examples can still exist.
> > > >
> > > > >The question “What resources (money, people, even electricity and water) are needed not only to get the first predictions but to annually update predictions with new training data? Is it really sustainable in the long term?” in Limitations is not related to our study. We did not claim that we had proposed a new and efficient method to analyze protein datasets. Could you explain what you meant by this question?
> > > >
> > > > After Geoffrey Hinton, Yoshua Bengio and others publicly accepted that AI is a risk to humanity, the panel discussions at CVPR 2023 and ICML 2023 raised serious concerns about using black-box tools without guarantees in life-critical scenarios including protein biology.
> > > >
> > > > Did the authors read the recommended paper "The impact of AlphaFold2 one year on" at https://www.nature.com/articles/s41592-021-01365-3, which explicitly asked whether machine learning approaches are justified in terms of required resources and lack of guarantees?

---

> > > > > ### Comment · Area_Chair_hx65 · 2023-08-18
> > > > >
> > > > > As your AC, I want to briefly provide a better understanding of the 'Limitations' section: the points raised by the reviewer would better be assessed by an additional *Ethics Review*. In my opinion as an AC, such an Ethics Review is **not warranted here**, though, because the authors use the proteins data set only as an example and make no claims about deploying their work in a real-world application.

---

> > > > > ### Author Response · Authors · 2023-08-21
> > > > >
> > > > > Thank you for your further questions.
> > > > >
> > > > > > Is this classification subjective, not based on a rigorously defined equivalence?
> > > > >
> > > > > > What problem did the paper aim to solve?
> > > > >
> > > > > As we stated in the contribution part of the paper, our study aims to propose a novel method that can improve classification accuracy. We work on the classification task for the point cloud data that are meaningfully labeled beforehand. This type of classification is not directly related to the rigorous equivalence of point clouds.
> > > > >
> > > > > > Have you checked that that the combined features distinguish all point clouds in your data?
> > > > >
> > > > > The networks in our method are trained to improve classification accuracy, so it does not make sense to verify whether data points belonging to different labels have different features after the training is completed. Therefore, we do not have to check them.
> > > > >
> > > > > > Then does it makes sense to compare your output with the past work on isometry invariants? Table 2 includes comparisons with the past work on Rips, DTM, DistMatrixNet. Is the output of DistMatrixNet invariant under permutation of points?
> > > > >
> > > > > > Are there any guarantees that Rips and DTM don't map non-equivalent inputs to the same output? Such guarantees where proved for the distribution of pairwise distances on almost all Euclidean clouds by Boutin and Kemper and Adv. Appl. Math. (2004).
> > > > >
> > > > > We conducted experiments to compare the classification accuracy of our method to Rips, DTM, and DistMatrixNet. The output of DistMatrixNet is invariant under the permutation of points as we described in Section 3.1 of the paper.
> > > > >
> > > > > There might be examples that non-isometric (or non-equivalent) point clouds whose persistent homology are the same. This is not directly relevant to the classification of labeled point clouds, since our aim is not to distinguish point clouds up to isometry (or equivalence relation).
> > > > >
> > > > > > This past work seems much stronger because the distribution of all pairwise distances needs only O(m^2) time on m unordered points, is Lipschitz continuous under perturbations of given and generically complete as proved nearly 20 years ago. Did you know about this work?
> > > > >
> > > > > The purpose of our method is to classify the labeled point clouds, which is different from that work. Rather, this study can help to improve our method. Thank you again for letting us know about this paper.
> > > > >
> > > > > > If this result is well-known, is it possible to give a reference?
> > > > >
> > > > > The complexity is cubic in the number of simplices. You can find this fact in, for example, the last sentence of Section 5.3.1 in Otter et al., 2017, “A Roadmap for the Computation of Persistent Homology”. See also Dmitriy Morozov, 2005, “Persistence Algorithm Takes Cubic Time in the Worst Case”.
> > > > >
> > > > > In our case, we need to consider up to 2-simplices, so the number of simplices is again cubic in the number of points N, which implies the whole complexity is O(N^9) in the worst case.
> > > > >
> > > > > > Does higher accuracy on a finite dataset guarantee a similarly high accuracy on other or larger datasets?
> > > > >
> > > > > We attempted to develop a method to achieve higher accuracy on classification tasks. We conducted experiments to showcase this method, and this is a standard way to demonstrate the efficacy of a proposed machine learning method. Such experiments do not guarantee high accuracy on other datasets in general, but we demonstrated the efficacy of our method with several types of datasets. Providing a rigorous guarantee for our method is beyond the scope of our paper.
> > > > >
> > > > > > After Geoffrey Hinton, Yoshua Bengio and others publicly accepted that AI is a risk to humanity, the panel discussions at CVPR 2023 and ICML 2023 raised serious concerns about using black-box tools without guarantees in life-critical scenarios including protein biology.
> > > > > Did the authors read the recommended paper "The impact of AlphaFold2 one year on", which explicitly asked whether machine learning approaches are justified in terms of required resources and lack of guarantees?
> > > > >
> > > > > As the AC mentioned, we used the protein dataset only as an example and we made no claims about deploying our work in a real-world application.

---

> > > > > > ### Comment · Reviewer_k7w9 · 2023-08-21
> > > > > > **types of proteins**
> > > > > >
> > > > > > Thank you for the reply.
> > > > > >
> > > > > > >we used the protein dataset only as an example and we made no claims about deploying our work in a real-world application.
> > > > > >
> > > > > > Yes, the area chair also raised the issue of "14 types of proteins". Could you please clarify in the previous reply "the protein dataset composed of 1,000 data by subsampling from 14 types of proteins", what is meant by a "type of a protein" and if "1,000 data" meant a single cloud of 1000 points or 1000 clouds (of how many points)? How are these points subsampled from a single protein? Thank you.

---

> > > > > > > ### Author Response · Authors · 2023-08-21
> > > > > > >
> > > > > > > Thank you for your question.
> > > > > > >
> > > > > > > > Could you please clarify in the previous reply “the protein dataset composed of 1,000 data by subsampling from 14 types of proteins”, what is meant by a “type of a protein” and if “1,000 data” meant a single cloud of 1000 points or 1000 clouds (of how many points)? How are these points subsampled from a single protein?
> > > > > > >
> > > > > > > As we explained in lines 266--276 of the paper, we classified 1000 point clouds, each containing 60 points. The original data, created by Kovacev-Nikolic et al. (2016), is 14 point clouds (one for each type of protein), each containing 370 points. Out of these point clouds, 7 belong to the "open", and the other 7 belong to the "close". For both of the two classes, we repeated the following 500 times: 1. randomly choose one of the 7 point clouds and 2. randomly subsample 60 points from that point cloud. As a result, we obtained 1000 point clouds, each containing 60 points.

---

> ### Author Response · Authors · 2023-08-10
>
> - The second to fourth paragraphs in weakness are not directly related to our paper. They seem just slander for the persistent homology community, which is based on the misunderstanding that “there was no high-level problem that persistent homology solves”.  Please see the first part of our rebuttal.
> - We did not really understand the sentence “Since the authors have written detailed and accurate mathematical proofs in the appendix, they could probably agree that *examples prove nothing* because counter-examples can still exist, especially when all possible data (as for point clouds) fill a continuous space.” in Limitations. Although we tried to answer this within our understanding, but could you explain it in more detail?
> - The question “What resources (money, people, even electricity and water) are needed not only to get the first predictions but to annually update predictions with new training data? Is it really sustainable in the long term?” in Limitations is not related to our study. We did not claim that we had proposed a new and efficient method to analyze protein datasets. Could you explain what you meant by this question?

---

### Official Review · Reviewer_L8GX · 2023-07-06

**Soundness:** 2 fair
**Presentation:** 3 good
**Contribution:** 3 good
**Rating:** 6
**Confidence:** 3

**Summary:**

A neural network that learns the filtration for persistent homology on given point cloud data is introduced, theoretically justified, and evaluated experimentally on 2 data sets.

**Strengths:**

(S1) If this is indeed the first work that considers learning filtrations on point clouds, I find the idea very relevant.

(S2) The filtration learning approach is very nicely motivated and described (Lines 185 – 196).

**Weaknesses:**

(W1) The need for learned filtrations should be better motivated. Think of an example point cloud where some other learnable filtration is more meaningful than Rips or DTM, visualize all three filtrations and their PDs. For example, we know that DTM is more suitable than Rips in the presence of outliers, but when is another filtration better than Rips and DTM? This would provide guidance to readers when it would make sense to use your approach, instead of relying simply on the Rips and DTM filtration. From Table 1 results, it seems that some answers might lie in the protein data set you consider.

(W2) [1] and [2] seem to be related work, but are not referenced?

(W3) The improvement with a learned filtration is good for protein classification, but it is for sure not convincing for the 3D CAD data (whereas it is much more complicated and computationally difficult). It would therefore be useful to provide more information about the data (including visualizations), and more detailed insights. I took a look at Appendix C, but this does not provide answers to these questions. Negative insights are also meaningful, i.e., that learning a filtration might not be useful for a lot of problems, and that relying on Rips or DTM filtration is good enough.

(W4) You write: “… the classification accuracy is better when our method was concatenated with DeepSets/PointNet compared to using DeepSets/PointNet alone. The accuracy when we combine our method and PointMLP is not higher than that of PointMLP. This would mean that concatenating the topological feature is effective when the number of parameters of a DNN-based method is relatively small.” I find this argument extremely flawed, since the number of parameters for PointNet and PointMLP is very comparable?


[1] Zhang, Simon, Soham Mukherjee, and Tamal K. Dey. "GEFL: Extended Filtration Learning for Graph Classification." Learning on Graphs Conference. PMLR, 2022.
[2] Horn, Max, et al. "Topological graph neural networks." arXiv preprint arXiv:2102.07835 (2021).


**Questions:**

(Q1) PH wrt some filtrations (e.g., height, useful to distinguish MNIST digit 6 from digit 9) is not isometry-invariant, so why do you impose this condition?

(Q2) Figure 1: Comment more why this approach is meaningful, since 2 holes are not recognized with 1-dim PD for any of the two point clouds? Consider plotting (next to) more reasonable weights.

(Q3) You never mention simplicial complexes, so that it remains unclear why the filtration discussed on lines 143-144 is Rips, and not Čech?

(Q4) How does your weighted Rips filtration compare to the weighted Rips filtration discussed in [3] (Proposition 3.5), where point cloud point x appears according to its filtration function value (weight) f(x), and an edge (x, y) appears when f(x), f(y) and distance d(x, y) satisfy certain properties?

(Q4) Related to (Q3) and (Q4), do all point cloud points immediately appear in your filtration? In particular, if a point has a really large weight, the balls centered at this point will expand very late in the filtration, but will the point (ball with radius 0) be there from the beginning? This is important e.g. if the point is an outlier, since we commonly want to ignore such a point.

(Q5) “Although (a) and (b) can be learned together since the output of the resulting feature is differentiable with all parameters, it would make the optimization unstable.” Why, can you explain more?

(Q6) In Section 5, can you provide more intuition on what is captured with DistMatrixNet?


Other minor comments:

-	The homology is persistent (not persistence homology), but we talk about persistence landscapes and images (not persistent landscape or image), rephrase throughout the paper.
-	Line 86: Explicitly mention DeepSets.
-	Line 142: “one can take function S defined by” -> “one can take S to be the distance to point cloud, defined by”
-	Line 156:  “which can also [add: be] computed only by distance matrices”. Do you not need the weights too?
-	Line 168: What is function u, where is it used?
-	Line 169: “we can vectorize persistent a persistent diagram”. Rephrase.
-	Line 234: Do topological features and DNN features have to have the same dimension L?
-	Line 251: “we can approximate any continuous function on X x [0,1]^m can be approximated”. Rephrase.
-	Line 253: Cite specific Appendix.
-	Line 283: “we replaced our topological featured replaced”. Rephrase.
-	Line 285: “Note that we use DistMatrixNet not for the computation of filtration weights.” This sentence seems weird, what do you mean?
-	Table 2 caption: “concatenated the feature” -> “concatenated with the feature”?
-	Mention explicitly that the code is made publicly available.
-	References: Check capitalization of acronyms in paper titles (e.g., Dtm, Perslay, Ripsnet, Toposeg, Homcloud, Pointnet, 3d, Pointet++, Pi-net, 3d shapenets, Sgmnet, …), and be consistent between capital case vs. lower case for journal names.

[3] Anai, Hirokazu, et al. "DTM-based filtrations." Topological Data Analysis: The Abel Symposium 2018. Springer International Publishing, 2020.


**Limitations:**

Some limitations and corresponding future research directions are mentioned, but they do not address the lack of insights on when a learned filtration can be expected to be beneficial (compared to e.g. Rips and DTM filtrations).

---

> ### Author Rebuttal · Authors · 2023-08-09
>
> We appreciate the detailed comments and suggestions for improving our paper. We will reflect all the comments and suggestions in our final version. In the following, we respond to specific concerns and questions raised by the reviewer.
>
> > (W1) The need for learned …
>
> Thank you for your important suggestion. To clarify the validity of learning filtration using our method, in Section 1, we will add a sentence “We will show some experimental results which show that our method improves the classification accuracy compared to when using Rips or DTM filtration".
>
> One example that motivates learning filtration is the point cloud in Figure A in the global response. In this example, the trained weight function gives large values to the outliers. Although this is similar to the DTM function, we remark that such a weight function was obtained in a data-driven and supervised way, without any information (other than data) in advance. We believe that the fact that our method can learn suitable filtrations for classification without any prior specification, as in this example, allows us to motivate the use of our method.
>
> > (W2) [1] and [2] seem …
>
> We are grateful to you for pointing out the lack of references. We will add these references in the final version.
>
> > (W3) The improvement with …
>
> Thank you for your constructive comments. We attached some visualization of the learned weights for the 3D CAD data (Figure B). We can observe that the points in some of the point clouds would have appropriate weight to be classified correctly, while there exist point clouds such that all points are assigned a weight of 0. This shows that learning filtration with our method would contribute to improving classification accuracy for some of the data, while is not effective for some data.
>
> On the other hand, as you pointed out, the accuracy improvement by the proposed method for 3D CAD data is not remarkable. Based on this, in the final version, we will describe that Rips or DTM filtration is effective enough for some data (such as the furniture surface data) so that our method does not lead to further improvements in accuracy while learning filtration with our method is beneficial for some data (such as protein data).
>
> > (W4) You write: “…” I find this …
>
> Thank you for your essential remark. After reviewing the results, we now believe that the lack of accuracy is due to incompatibility with PointMLP, rather than to the large number of parameters. We hypothesize that PointMLP has already captured enough information, including topological features, during the 1st phase. If so, the information obtained by persistent homology may be redundant, potentially negatively impacting the classification. We will appropriately replace the current observation with the hypothesis above in the final version.
>
> > (Q1) PH wrt some …
>
> In this paper, we focus on the classification of point clouds. In this setting, it would be natural to impose the isometry-invariance.
>
> > (Q2) Figure 1: Comment more …
>
> Thank you for your great idea on the image explaining the procedure of our method. We will replace Figure 1 with Figure A in the PDF as we stated in the global response.
>
> > (Q3) You never mention …
>
> > (Q4) How does your …
>
> > Related to (Q3) and (Q4), …
>
> We appreciate your comments that we should add some explanation about simplicial complex and Vietoris-Rips filtration. Due to the page constraints, we could not include them in the main text. We instead described the increasing family of balls, which gives an intrinsic understanding, but it was not an honest way. We will modify the sentence from line 145 and create a new section to give detailed explanations of simplicial complexes, Cech, and Rips filtrations in the appendix. More concretely, we will change the sentences from line 145 as follows: “The persistent homology of this filtration can be captured by the filtration called Cech or Vietoris-Rips filtration (Rips filtration for short). In this paper, we use Rips filtration for computational efficiency. See Appendix … for details”. In the same section in the appendix, we will also describe the weighted Rips filtration we used, which is totally the same as the one defined in Section 3.3 of Anai et al. (2020).
>
> > (Q5) “Although …” Why, can you explain more?
>
> The loss function can be differentiable with respect to all of the parameters in the network (a) and (b), but is not smooth with respect to the parameters in (a)*.* Because of this, if we optimize both of the networks in (a) and (b), not only the resulting networks of (a) but also that of (b) are (empirically) not optimized well. This is why we separately optimized the networks in (a) and (b). We will append these details in the final version.
>
> > (Q6) In Section 5,  …
>
> We believe that DistMatrixNet can extract the information of point clouds using the relative distance information. The experimental results show that DistMatrixNet is effective when used as a weight function in the proposed method, but not when used directly for classification. This might mean that DistMatrixNet can be effectively used to obtain information about the role of each point in the point cloud, but is not suitable for getting information to distinguish its global shape.  Further investigation on the role of the DistMatrixNet is future work.
>
> > Other minor comments:
>
> We appreciate your detailed comments and giving us the suggested fixes. We will fix every mistake in the final version.
>
> > **Limitations:** … they do not address …
>
> Thank you for your important remarks that the description in the limitation section is not enough. In the final version, we will add some insights on the learned filtration compared to Rips and DTM filtration, including negative ones.
>
> We hope that we addressed all your questions and concerns adequately. In light of our clarifications, please consider increasing your score to accept. Please let us know if we can provide any further details and/or clarifications.

---

> > ### Comment · Reviewer_L8GX · 2023-08-15
> >
> > Thank you for the detailed response. I am happy to see that you agree with the most of the suggestions, but I do worry whether and to what extent will the improvements will be included in the revised version of the manuscript, since the new general .pdf consists only of two additional figures.
> >
> > Some final comments:
> >
> > - I don’t think you answered (Q4), about the filtration function value on the simplices that are vertices?
> >
> > - I definitely agree that Figure A much better describes your motivation than the previous Figure 1. Reference the specific plots (a)-(d) in the caption, or remove the labels.
> >
> > - Figure B is very nice too. I wonder though, it would be much more interesting to see similar plots for the protein data where learning the filtration is yielding better results than Rips and DTM? It would be particularly nice to also add plots here with the value of the DTM filtration function on the point cloud points (the average distance from a number of nearest neighbors), to see how your learned weights differ from the DTM filtration, and gain some insights on what was essential for the good performance of your method. Plotting some sets in the filtration, and the resulting persistence diagrams, for some interesting point clouds (where the learned weight is different from DTM) would also be very interesting to see.
> >
> > - Why do you include the first row in Figure B, where the color represents the x coordinate, what information does this give us? For the second row, I don’t understand how do you see that “points in some of the point clouds would have appropriate weight to be classified correctly”? You should also include the color legends in both figures, to make it clear which points end up having lower weight.
> >
> > - I would suggest not undermining your method too much, and rewrite “while is not effective for some data” in the Figure B caption to “while the learning of the filtration does not have an added value compared to the standard Rips filtration for some other data”, also to improve clarity.
> >
> > - It would be interesting to try to gain some understanding on *why* learning the filtration is beneficial for the protein data (and not for the furniture data). Experiments on more data sets would be very helpful in this direction. In any case, visualizing the two data sets has now become even more important (e.g., at least by including an analogous figure to Figure B for the protein data, as already suggested above).
> >
> > - To improve the impact of the paper, I would suggest to include a small Jupyter notebook as a part of your code, allowing the user to visualize the learned weights for their problem at hand. This will not influence my final rating of the paper, and is obviously up to you to decide if it makes sense.

---

> > > ### Author Response · Authors · 2023-08-18
> > >
> > > We appreciate your beneficial comments.
> > >
> > > > I don’t think you answered (Q4), about the filtration function value on the simplices that are vertices?
> > >
> > > Each vertex appears at $t=f(x)$, so they are not present at the beginning if the weights are greater than zero. This is the same as the weighted Rips filtration that is defined in Section 3.3 of Anai et al. (2020).
> > >
> > > > I definitely agree that Figure A much better describes your motivation than the previous Figure 1. Reference the specific plots (a)-(d) in the caption, or remove the labels.
> > >
> > > Thank you for your favorable feedback on Figure A. We will add the precise description in the caption with referring (a)-(d). (We omitted some of the descriptions in the caption that are the same as original Figure 1. )
> > >
> > > > I wonder though, it would be much more interesting to see similar plots for the protein data where learning the filtration is yielding better results than Rips and DTM? It would be particularly nice to also add plots here with the value of the DTM filtration function on the point cloud points (the average distance from a number of nearest neighbors), to see how your learned weights differ from the DTM filtration, and gain some insights on what was essential for the good performance of your method. Plotting some sets in the filtration, and the resulting persistence diagrams, for some interesting point clouds (where the learned weight is different from DTM) would also be very interesting to see.
> > >
> > > > It would be interesting to try to gain some understanding on why learning the filtration is beneficial for the protein data (and not for the furniture data). Experiments on more data sets would be very helpful in this direction. In any case, visualizing the two data sets has now become even more important (e.g., at least by including an analogous figure to Figure B for the protein data, as already suggested above).
> > >
> > > Thank you for your constructive comments. We will additionally visualize the weight function for the protein dataset, compare them to the DTM filtration and observe the persistence diagrams for some of the point clouds to demonstrate the effectiveness of our method.
> > > We will include these results in the paper as much as the page constraints. (If there is not enough space, we will add them to the Appendix.)
> > >
> > > > Why do you include the first row in Figure B, where the color represents the x coordinate, what information does this give us? For the second row, I don’t understand how do you see that “points in some of the point clouds would have appropriate weight to be classified correctly”? You should also include the color legends in both figures, to make it clear which points end up having lower weight.
> > >
> > > We appreciate your important questions and comments.
> > >
> > > We included a colored point cloud based on the x-coordinates in order to enhance the visibility of the surface shape when plotting a 3D point cloud on a 2D plane.
> > >
> > > Regarding the weight function learned by our method, for example, in the rightmost point cloud, it appears that holes are formed at the upper and lower parts of the point cloud by the points with smaller weights.
> > > It might not be clear how this weight function is effective in the classification from these figures, so we will also present the associated persistence diagrams.
> > >
> > > Furthermore, we will include the color legends, as you suggested.
> > >
> > > > I would suggest not undermining your method too much, and rewrite “while is not effective for some data” in the Figure B caption to “while the learning of the filtration does not have an added value compared to the standard Rips filtration for some other data”, also to improve clarity.
> > >
> > > Thank you for your helpful comment. We will replace the expression in the caption of Figure B.
> > >
> > > > To improve the impact of the paper, I would suggest to include a small Jupyter notebook as a part of your code, allowing the user to visualize the learned weights for their problem at hand. This will not influence my final rating of the paper, and is obviously up to you to decide if it makes sense.
> > >
> > > Thank you for your constructive suggestion.
> > > We will publish the source code that we used to visualize the weights of each point as a Jupyter notebook.

---

### Official Review · Reviewer_kFjr · 2023-07-07

**Soundness:** 2 fair
**Presentation:** 3 good
**Contribution:** 3 good
**Rating:** 5
**Confidence:** 3

**Summary:**

This paper investigates the extraction of global topological features using the framework of persistent homology. The authors have proposed a neural network architecture to learn the filtration weights for each point in an end-to-end and data-driven manner, which is later supported by an approximability theorem. Additionally, a two-phase training procedure is introduced to further improve the performance of the proposed architecture. The proposed framework is then applied to different tasks such as protein 3D CAD classifications.


**Strengths:**

1. [Originality] The filtration weight is usually constructed without considering the label information, e.g., a constant for VR complex and k-NN info for DTM. The authors proposed a framework to learn the weights for each point from the label using a neural network.
1. [Quality] Being able to provide an approximability theorem to support/motivate the choice of the neural network architecture is nice.
1. [Clarity] Great overview on the persistent homology as well as the limitations of the current framework. The author laid out the contribution in the very beginning of the introduction which can help present the novelty of this work. Overall, a well-written paper.
1. [Significance] TDA framework has long been suffered from the topological noise issue. Being able to propose a way to automatically learn a weight function to suppress the noises is an important contribution to the field.


**Weaknesses:**

1. It looks like there are two contributions in this paper: 1) propose a way to learn the weighted filtration as per Section 3.1, with theoretical guarantee in Section 4; and 2) Section 3.2 in the ability to approximation with composite function. The contribution in 1) is more related to filtration learning as it learns a weighting function, but 2) is a more general use case. However, by looking at the experimental results, it is not clear which contribution is more significant, i.e., the performance gain in Table 1 comes from 1), but the gain in Table 2 comes from 2). It would be nice to consolidate the comparisons between different experiments and provide some discussions. That way, the readers can get a better understanding of the interplay between different contributions.
1. It is not clear to me from reading the main text on how you choose the hyper-parameters for DTM. From reading the appendix, it looks like the parameters are not chosen by cross-validation (Section C.2). Is there any specific reason to not choose this parameter end-to-end using cross-validation? How will a different $q$ and $k$ affect the prediction performance in the protein datasets?
1. It looks like Theorem 4.1 suggests the function can be approximated, but it does not mention whether it can be recovered. Will it be possible to get some results regarding the convergence (i.e., with $n \to \infty$, will the $\epsilon$ shrink?) and/or whether $\psi_1$ and $\psi_2$ can be estimated by the proposed architecture (i.e., how close $\psi_1$ and $\psi_2$ to $h$ and $\phi^{(6)}$, respectively).
1. Related to #4, can we support Theorem 4.1 by running a synthetic example? Specifically, can we show that the true weighting function $f$ can be recovered by the $f_\theta$ in Figure 2?


**Questions:**

1. TDA is also used as data-analysis/unsupervised purposes (e.g., in finding enclosing holes) [A-C], I am curious whether the learned filtration weight $f(X, \cdot)$ can reveal the true topological structures or if one can design some sort of loss function and learn the weight function accordingly?
1. How to correctly understand the connection between the architecture (in Figure 2) and Theorem 4.1? The weighting function $f_\theta(X, x_1)$ is a concatenation of $h(X)$ and $g_1(x)$, but in Theorem 4.1, the $f(X, x)$ can be approximated by a concatenation of $\varphi_1(X)$ and $x$ itself.
1. [Minor language issue] When I first read the paper, it is not clear what the “architecture” and “approximation result” in L12-13 meaning (original sentence: “Additionally, we theoretically show a finite-dimensional approximation result that justifies our architecture.”). Consider adding some details there to improve clarification. For instance, you might want to change it to something like this: “Additionally, we theoretically show a finite-dimensional approximation \textbf{of any filtration function}, which \textbf{justifies (or motivates) the proposed neural network architecture}.”

---
[A] Wasserman, Larry. “Topological Data Analysis.” Annual Review of Statistics and Its Application 5 (2018): 501–32.

[B] Chen, Yu-Chia, and Marina Meila. “The Decomposition of the Higher-Order Homology Embedding Constructed from the k-Laplacian.” Advances in Neural Information Processing Systems 34 (2021).

[C] Wu, Pengxiang, Chao Chen, Yusu Wang, Shaoting Zhang, Changhe Yuan, Zhen Qian, Dimitris Metaxas, and Leon Axel. “Optimal Topological Cycles and Their Application in Cardiac Trabeculae Restoration.” In International Conference on Information Processing in Medical Imaging, 80–92. Springer, 2017.



**Limitations:**

Authors have addressed the limitations of their work. Negative social impact statement is not necessary in this work, as the primary focus of this manuscript lies in its theoretical contribution.

---

> ### Author Rebuttal · Authors · 2023-08-09
>
> We are grateful for the detailed comments and suggestions for improving our paper. We will reflect all the comments and suggestions in our final version. In the following, we respond to specific concerns and questions raised by the reviewer.
>
> > Weakness 1. It looks like there are two contributions in this paper: 1) propose a way to learn the weighted filtration as per Section 3.1, with theoretical guarantee in Section 4; and 2) Section 3.2 in the ability to approximation with composite function. The contribution in 1) is more related to filtration learning as it learns a weighting function, but 2) is a more general use case. However, by looking at the experimental results, it is not clear which contribution is more significant, ...
>
> Section 3.2 describes how to combine topological features with more general features obtained by DNNs to solve classification tasks, and it does not describe the ability to approximation with composite function. The first experiment in Section 5 was conducted to show the validity of our method where we only used the topological feature, without DNN features. In the second experiment, on the other hand, we combine the topological features and features from DNNs using the method described in Section 3.2. In the final version, we will clearly state the difference between the two experiments at the beginning of Section 5.
>
> > Weakness 2. It is not clear to me from reading the main text on how you choose the hyper-parameters for DTM. From reading the appendix, it looks like the parameters are not chosen by cross-validation (Section C.2). ...
>
> Thank you for your important remark. As you pointed out, the experimental results in the main text were not ones selected in cross-validation, while we conducted experiments with some hyper-parameters as we showed in Appendix. In the final version, we would like to fix them as we showed in the global response.
>
> > Weakness 3. It looks like Theorem 4.1 suggests the function can be approximated, but it does not mention whether it can be recovered. Will it be possible to get some results regarding the convergence ... and/or whether $\psi_1$ and $\psi_2$ can be estimated by the proposed architecture...
>
> > Question 2. How to correctly understand the connection between the architecture (in Figure 2) and Theorem 4.1? ...
>
> Thank you for your essential remark. Since we proved Theorem 4.1 to demonstrate the validity of our idea to concatenate the (finite-dimensional) global feature $\psi_1(X)$ (which corresponds to $h(X)$) and local feature $x$ (which corresponds to $g(x)$), it does not clarify whether  $\psi_1$ or $\psi_2$ can be recovered by the network architecture proposed in this paper indeed. We leave it as future work to provide stronger theoretical results for our network or to propose a new architecture with a complete approximation guarantee.
>
> > Weakness 4. Related to #4, can we support Theorem 4.1 by running a synthetic example? Specifically, can we show that the true weighting function $f$ can be recovered by the $f_\theta$ in Figure 2?
>
> We appreciate your constructive suggestion that we should support Theorem 4.1 by running a synthetic example.
>
> In this setting, we do not have a “true weight function”, but we just find the filtration that can achieve high classification accuracy.
>
> We have the experimental result to support Theorem 4.1 that shows our network can recover the DTM function, as shown in the table below. This table shows the error when our network is trained by the regression task to approximate the DTM functions. This result means that our method can choose filtrations from the space including Rips and DTM filtration if trained appropriately. We will add this result in the final version.
>
> | value of k |      error      |
> |:----------:|:---------------:|
> |  0 (Rips)  | 0.0000 ± 0.0000 |
> |      2     | 0.0015 ± 0.0000 |
> |      3     | 0.0016 ± 0.0000 |
> |      4     | 0.0017 ± 0.0000 |
> |      5     | 0.0018 ± 0.0000 |
> |     10     | 0.0022 ± 0.0001 |
>
> > Question 1. TDA is also used as data-analysis/unsupervised purposes (e.g., in finding enclosing holes) [A-C], I am curious whether the learned filtration weight $f_\theta(X, \cdot)$ can reveal the true topological structures or ...
>
> Thank you for your interesting comments. We believe that we can sometimes get some insights on the topological structure of the point cloud data $X$ from the learned filtration weight $f_\theta(X, \cdot)$. For instance, for the point clouds shown in Figure A (in the PDF of the global response), one can find that the points with large weights are outliers. We remark that these weights were automatically learned in a data-driven way without any prior information.
>
> As for the loss function, we are currently using the classification loss as a loss function, but we can also consider other types of loss functions.
> For example, if we consider the reconstruction task and its loss, one might obtain the filtration weight that is effective in extracting all of the topological information in the point clouds.
> In fact, we have conducted such an experiment and obtained appropriate weights in some cases. We leave it as future work to do much investigation on how using other types of loss functions affects the filtration weights.
>
> > Question 3. [Minor language issue] When I first read the paper, it is not clear what the “architecture” and “approximation result” in L12-13 meaning … Consider adding some details there to improve clarification. For instance, …
>
> Thank you for pointing out this issue and suggesting an alternative sentence. In the final version, we will change the expression to clarify how our theoretical result contributes.
>
> We hope that we addressed all your questions and concerns adequately. In light of our clarifications, please consider increasing your score to accept. Please let us know if we can provide any further details and/or clarifications.

---

> > ### Comment · Reviewer_kFjr · 2023-08-17
> >
> > Thank you for your response, which answers some of the questions I have! Given that TDA/PH is also used in data analysis part, I will highly suggest you to discuss those in your final version of the paper.
> >
> > I think the paper is still borderline, but I am leaning toward accept the paper now. I raised my score to 5 as a result.

---

> > > ### Author Response · Authors · 2023-08-18
> > >
> > > Thank you for your suggestion. We will include the discussion in the final version.

---

### Official Review · Reviewer_xfZT · 2023-07-08

**Soundness:** 2 fair
**Presentation:** 3 good
**Contribution:** 2 fair
**Rating:** 5
**Confidence:** 3

**Summary:**

This work proposes a novel framework to obtain adaptive topological features for point clouds based on persistent homology by introducing an isometry-invariant network architecture for a weight function and proposes a way to learn a weighted filtration. The work theoretically proves that any continuous weight function can be approximated by the composite of two continuous functions which factor through a finite-dimensional space. The experiments on public datasets shows the proposed method improves the accuracy in classification tasks.

All of my questions are carefully addressed in the rebuttal. The work is theoretically solid and has potential.

**Strengths:**

The idea of learning filtration by the weight function is novel. The approximation ability theorem is important. The architecture design is based on this theorem and the isometry-invariance, which is rigorous and interpretable. The experimental results are convincing.  The paper is well written, all the concepts, math symbols, theorems are thoroughly explained. The deductions are clear and easy to follow. The experimental results are convincing.

**Weaknesses:**

It is not clear why different weight functions affect the qualities of the topological features, and what are the criteria for the persistent diagrams. Some theoretical explanations and numerical demonstration will be helpful.

**Questions:**

1. What are the criteria for good weight functions ? What are the theoretical explanations for them ?
2. For the purpose of classification, the extracted topological features improves the results. It is unclear how the loss functions are differentiable with respect to the topological features and in turn the weight functions. The homology is intrinsically discrete, the differentiability is not obvious. Some explanation will be helpful.
3. Are the weight functions affected by the point cloud quality ?  For example, if the scanning quality is improved, how does that affect the weight function?

**Limitations:**

The work can be further improved by use more realistic examples and compare with state-of-the-art models.

---

> ### Author Rebuttal · Authors · 2023-08-09
>
> We appreciate the valuable feedback for improving our paper. We will reflect all the comments and suggestions in our final version. In the following, we respond to specific concerns and questions raised by the reviewer.
>
> > It is not clear why different weight functions affect the qualities of the topological features, and what are the criteria for the persistent diagrams. Some theoretical explanations and numerical demonstration will be helpful.
>
> > Question 1. What are the criteria for good weight functions? What are the theoretical explanations for them?
>
> The persistent homology depends on the weight function; Rips filtration is the case when the weights are all zero, and some studies(, for example, Anai et al. (2020)) have used the DTM function to deal with outliers. In this study, we took the approach of learning these weights in a data-driven and supervised manner in order to increase classification accuracy. Therefore, we can say that the classification accuracy is a criterion for the weights in this case.
>
> > Question 2. For the purpose of classification, the extracted topological features improves the results. It is unclear how the loss functions are differentiable with respect to the topological features and in turn the weight functions. The homology is intrinsically discrete, the differentiability is not obvious. Some explanation will be helpful.
>
> Thank you for your important suggestion. Although the homology is intrinsically discrete, the differentiability of persistent homology has already been discussed in previous studies such as [1], [2], [3], and [4]. The differentiability of the loss function in our method can be directly derived from these results. We believe that such a differentiability argument is now standard and it would not be needed in the main text. We would add such an argument in the appendix if necessary.
>
> [1] M. Gameiro et al. A topological measurement of protein compressibility. *Japan Journal of Industrial and Applied Mathematics*, 32:1–17, 2015.
>
> [2] C. Chen et al. A topological regularizer for classifiers via persistent homology. In *The 22nd International Conference on Artificial Intelligence and Statistics*, pages 2573–2582. PMLR, 2019.
>
> [3] M. Carrière et al. Optimizing persistent homology based functions. In *International Conference on Machine Learning*, pages 1294–1303. PMLR, 2021.
>
> [4] J. Leygonie et al. A framework for differential calculus on persistence barcodes. *Foundations of Computational Mathematics*, pages 1–63, 2021.
>
> > Question 3. Are the weight functions affected by the point cloud quality? For example, if the scanning quality is improved, how does that affect the weight function?
>
> We are grateful for your intriguing question. We believe that the weight functions would be affected by the quality of the point clouds. However, we are currently not sure how the affection will be. To clarify this, in our future work, we will conduct the experiment to investigate how our trained filtration will change if the scale of the noise (for example, the variance of Gaussian noise) in the dataset is changed.
>
> We hope that we addressed all your questions and concerns adequately. In light of our clarifications, please consider increasing your score to accept. Please let us know if
> we can provide any further details and/or clarifications.

---

> ### Comment · Area_Chair_hx65 · 2023-08-18
>
> Dear reviewer,
>
> Please **briefly acknowledge the rebuttal** by the authors and consider updating your score—we want to avoid borderline scores for reviews, and the discussion phase will close soon. If you have any additional questions to the authors  please ask them **now**.
>
> Thanks,\
> Your AC

---

### Author Rebuttal · Authors · 2023-08-09

We appreciate the detailed comments and suggestions for improving our paper. We will reflect all the comments and suggestions in our final version. In the following, we show you some information that we would like to share with all of the reviewers.

1. A reviewer pointed out that Figure 1 is confusing. To address this issue, we would like to replace Figure 1 with Figure A in the attached PDF.
2. A reviewer pointed out that the hyper-parameter $k$ and $q$ in DTM filtration should be chosen to maximize the average classification accuracy with cross-validation. While the experimental results for DTM with different parameters were included in the Appendix, the results in the main text were not for optimal parameters, so we will replace this with the following table in the final version.

    For protein data:

    | DistMatrixNet | Rips | DTM | Ours |
    | --- | --- | --- | --- |
    | 65.0 ± 12.0 | 79.9 ± 3.0 | 78.0 ± 1.6 | 81.9 ± 2.1 |

    For 3D CAD data:

    |  |  | DeepSets | PointNet | PointMLP |
    | --- | --- | --- | --- | --- |
    | 1st Phase |  | 65.7 ± 1.4 | 64.3 ± 4.4 | 68.8 ± 6.3 |
    | 2nd Phase | DistMatrixNet | 65.7 ± 4.8 | 55.7 ± 13.9 | 53.8 ± 7.4 |
    |  | Rips | 67.0 ± 2.6 | 68.4 ± 2.4 | 57.8 ± 12.4 |
    |  | DTM | 68.0 ± 2.5 | 68.7 ± 2.3 | 57.2 ± 6.8 |
    |  | Ours | 67.5 ± 2.5 | 68.8 ± 2.0 | 60.0 ± 6.3 |
3. A reviewer pointed out we should visualize the resulting weight function learned by our method. We showed some examples of the weight function for 3D CAD data in Figure B in the attached PDF. We will add this figure in the appendix.

We also respond to the comments from each reviewer. If there are further questions/comments/suggestions, we would be happy to address them in the discussion period.

---

> ### Comment · Reviewer_L8GX · 2023-08-11
> **DTM > Rips**
>
> Regarding your Comment 2: If the parameters of the DTM filtration are tuned properly, it should always obtain at least as good performance as Rips, for k = 0 and q = 2. Why is DTM underperforming for the protein data in your first table of results above, maybe you do not consider these parameter values? In any case, please make sure to explicitly specify the list of considered values for k and q.

---

> > ### Author Response · Authors · 2023-08-12
> >
> > Thank you for your feedback.
> > We have chosen the hyperparameter k for the DTM filtration from among 2, 3, 5, 8, and 10.
> > In this experiment, we fixed the value of q at 2.
> > We consider the Rips filtration (corresponding to k = 0) to be the most commonly used one, and we presented its results separately.
> > We will include this information in the final version.

---

### Decision · Program_Chairs · 2023-09-21

**Decision:**

Accept (poster)

**Comment:**

Reviewers agreed that the current submission constitutes a relevant and timely addition to the ever-growing literature of persistent homology in general, and topological machine learning in particular. While some concerns were raised about some methodological details, the authors provided a strong rebuttal and discussed all aspects satisfactorily.

I am thus happy to endorse this paper for publication, primarily because the learning of filtration for point clouds proves to be a **substantial algorithmic improvement** for computational topology and topological machine learning. While some works already learn [one](https://proceedings.mlr.press/v119/hofer20b) or [more](https://openreview.net/forum?id=oxxUMeFwEHd) filtrations in the graph context, current work for point clouds is mostly concerned with defining new filtrations, without having a way to adjust them to the task at hand. This critical need is now addressed by the paper at hand. Using an elegant and extensible architecture, the proposed work constitutes an important framework that will serve as the basis for many future extensions. A particularly relevant aspect for learning filtrations, also missing from previous work, constitutes an analysis of the expressive power; the authors close this gap and show that their architecture can approximate all relevant weight functions.

 The concerns by reviewer `k7w9` are acknowledged, but pertain to the wider area of persistent homology, not this specific paper. The authors are **strongly encouraged** to incorporate the comments made by the reviewer, in particular those pertaining to the experimental setup—additional data sets that showcase the power of the proposed method would strengthen the paper more. Likewise, additional work concerning clarity improvements would help to make the better overall more accessible and increase its well-deserved visibility.